# Antimicrobial Activities of Highly Bioavailable Organic Salts and Ionic Liquids from Fluoroquinolones

**DOI:** 10.3390/pharmaceutics12080694

**Published:** 2020-07-23

**Authors:** Miguel M. Santos, Celso Alves, Joana Silva, Catarina Florindo, Alexandra Costa, Željko Petrovski, Isabel M. Marrucho, Rui Pedrosa, Luís C. Branco

**Affiliations:** 1LAQV-REQUIMTE, Departamento de Química, Faculdade de Ciências e Tecnologia, Universidade Nova de Lisboa, 2829-516 Caparica, Portugal; amp.costa@campus.fct.unl.pt (A.C.); z.petrovski@fct.unl.pt (Ž.P.); 2MARE–Marine and Environmental Sciences Centre, ESTM, Instituto Politécnico de Leiria, 2520-641 Peniche, Portugal; celso.alves@ipleiria.pt (C.A.); joana.m.silva@ipleiria.pt (J.S.); 3Centro de Química Estrutural, Instituto Superior Técnico, Universidade de Lisboa, Avenida Rovisco Pais, 1049-001 Lisboa, Portugal; catarina_florindo@hotmail.com (C.F.); isabel.marrucho@tecnico.ulisboa.pt (I.M.M.)

**Keywords:** active pharmaceutical ingredients as organic salts and ionic liquids (API–OSILs), antibiotics, ciprofloxacin, fluoroquinolones, ionic liquids, norfloxacin, polymorphism

## Abstract

As the development of novel antibiotics has been at a halt for several decades, chemically enhancing existing drugs is a very promising approach to drug development. Herein, we report the preparation of twelve organic salts and ionic liquids (OSILs) from ciprofloxacin and norfloxacin as anions with enhanced antimicrobial activity. Each one of the fluoroquinolones (FQs) was combined with six different organic hydroxide cations in 93–100% yield through a buffer-assisted neutralization methodology. Six of those were isomorphous salts while the remaining six were ionic liquids, with four of them being room temperature ionic liquids. The prepared compounds were not toxic to healthy cell lines and displayed between 47- and 1416-fold more solubility in water at 25 and 37 °C than the original drugs, with the exception of the ones containing the cetylpyridinium cation. In general, the antimicrobial activity against *Klebsiella pneumoniae* was particularly enhanced for the ciprofloxacin-based OSILs, with up to ca. 20-fold decreases of the inhibitory concentrations in relation to the parent drug, while activity against *Staphylococcus aureus* and the commensal *Bacillus subtilis* strain was often reduced. Depending on the cation–drug combination, broad-spectrum or strain-specific antibiotic salts were achieved, potentially leading to the future development of highly bioavailable and safe antimicrobial ionic formulations.

## 1. Introduction

Fluoroquinolones are a class of antibiotics primarily effective against Gram-negative bacteria. Norfloxacin was the first fluoroquinolone to receive US Food and Drug Administration approval and is currently used to treat urinary, biliary, and respiratory tract infections [1]. One year later, in 1987, ciprofloxacin was also approved for use in the US, and it was the most successful compound in its generation as it showed 4–10 times more efficiency than its predecessor [2]. This drug is very effective in the treatment of a wide range of infections such as urinary tract infection, osteomyelitis (bone infection), respiratory infections and some sexually-transmitted diseases (e.g., gonococcal and chronic bacterial prostatitis) [3].

Fluoroquinolones, like many other active pharmaceutical ingredients (APIs), display spontaneous polymorphic conversion between distinct crystalline forms with a consequent change in pharmaceutical properties [4]. Crystal structures of neutral and zwitterionic forms of norfloxacin and ciprofloxacin have been described by Nikaido and Thanassi, with considerable differences in their bioavailability [5].

In aqueous solutions, a dynamic equilibrium of several protolytic forms (anionic, neutral, zwitterionic and cationic) can be found for both antibiotics due to the presence of distinct functional groups that are prone to converse protonation/deprotonation in physiological media. Such behaviour clearly influences their bioavailability, with neutral ciprofloxacin and norfloxacin displaying 0.086 and 0.37 mg/mL of solubility in water at 25 °C, respectively [6]. The typical conversion to the halide salt renders a considerable increase of two orders of magnitude in water solubility, with the ciprofloxacin hydrochloride salt reaching 38.4 mg/mL at 30 °C [7]. However, it is still considered a poorly water soluble drug.

According to the guidelines of the United States Pharmacopeia [8], solubility improvement techniques of poorly water soluble active pharmaceutical ingredients can be categorised into physical modification, such as chemical modifications of the drug substance, and other techniques. The formation of organic salts of a drug is a mature and explored chemical modification technique.

For more than a decade, active pharmaceutical ingredients as organic salts and ionic liquids (API–OSILs) have risen in academia as an alternative formulation for low bioavailable drugs [9,10,11,12]. This third generation of ionic liquids [9] consists on the combination of APIs as cations or anions with organic counterions, thereby inducing distinctive physicochemical properties over the original drugs and reduced toxicity to healthy cells, thus rendering a potentially enhanced pharmaceutical activity in comparison to the API [13,14,15,16,17,18,19,20,21,22,23,24,25]. Our works involving the preparation of API–OSILs from, e.g., β-lactam (ampicillin [18,19,20,21], penicillin [22] and amoxicillin [22]) and fluoroquinolone [18,23] antibiotics, NSAIDs (ibuprofen [17,18], naproxen [18]), bone antiresorptive agents (zoledronic [24] and alendronic [25] acids), among others, have shown that the combination of an API, either as a cation or as an anion, with suitable biocompatible counter-ions can increase the water solubility of the parent drug and even change its biological effect [10,17,23,24,25,26,27,28]. Furthermore, the stability and solubility in physiological media of the API can be significantly altered, yielding novel formulations with different pharmacokinetics and potentially different delivery modes and applications [29,30]. In addition, the polymorphic tendency of an API can be considerably reduced or even eliminated (e.g., by attaining the liquid state at room temperature), hence tackling one of the most important issues in the pharmaceutical industry, which is responsible for limitations in a drug’s solubility and dosage [17,23,24].

The formation of organic salts based on cationic fluoroquinolones (mostly comprising ciprofloxacin (Cip), but also norfloxacin (Nor)) has been widely explored by the academic community by protonation of the drugs’ amine group(s). More precisely, these fluoroquinolones have been combined with dicarboxylic acids (malonic and tartaric [31], adipic, fumaric [32] and maleic [33,34], succinic [35]), citric acid [32], saccharine [36,37] and acesulfame [36], as well as with other drugs (diflunisal and ibuprofen [38], salicylic [39] and barbituric acids [40]). Some works involve the combination of ciprofloxacin or enrofloxacin as cations and/or as anions with amino acids [41] and ionic polymers [42,43]. While most works focus on the study of the chemical and physicochemical properties of the prepared salts, including bioavailability, crystallinity/amorphous profiles and thermodynamic stability, the amorphous–solid dispersions of ciprofloxacin and enrofloxacin showed, in addition, that the antimicrobial activity of the salts is enhanced in comparison with the parent drugs, in particular against Gram-negative bacteria strains.

In addition to these works, our group reported the synthesis and characterization of novel API–OSILs from Cip and Nor as cations in combination with methanesulfonate (Mes), gluconate (Glu) and glycolate (Gly) biocompatible anions [23] (see Figure 1). The bioavailability of the prepared fluoroquinolone-based salts (FQ–OSILs), expressed in terms of permeability (octanol/water and phospholipid/water partition) and solubility in water and biological fluids, was highly tuneable according to the selected anion. Finally, toxicological studies on intestinal epithelial cell line models (Caco-2) revealed a lower inflammatory response for some of the FQ–OSILs in comparison with the parent drugs.

In this work, we set out to explore for the first time the formation of FQ–OSILs based on Cip and Nor as anions in combination with organic cation hydroxides [17,42] based on ammonium, pyridinium and *N*-methylimidazolium moieties (Figure 1).

The newly synthesized FQ–OSILs were characterised by nuclear magnetic resonance (NMR) (^1^H and ^13^C) and Fourier-transform infrared (FTIR) spectroscopic techniques, elemental analysis, as well as differential scanning calorimetry (DSC). Water solubility at 25 and 37 °C was experimentally determined for the FQ–OSILs, with the exception of the ones containing the [C_16_Py] cation, for which the critical micelle concentration was measured. Finally, the antimicrobial activity of these anionic FQ–OSILs on Gram-negative (*Klebsiella pneumoniae*) and Gram-positive (*Bacillus subtilis*, *Staphylococcus aureus*) bacteria was determined.

## 2. Results and Discussion

### 2.1. Synthesis

Ciprofloxacin and norfloxacin were combined as anions with choline [Ch], 1-ethyl-3-methylimidazolium [EMIM], 1-hydroxy-ethyl-3-methylimidazolium [C_2_OHMIM], 1-(2-hydroxyethyl)-2,3-dimethylimidazolium [C_2_OHDMIM], 1-(2-methoxyethyl)-3-methylimidazolium [C_3_OMIM] and cetylpyridinium [C_16_Py] cations (see Scheme 1).

According to our experience [17,24], the most efficient methodology for the combination of APIs as anions with organic cations consists of the dropwise addition of the hydroxide salts of the cations to the neutral API, leading to the sole formation of the desired API–OSILs and water. The cation hydroxides are prepared immediately before their addition from the corresponding halide salts by anion exchange from a hydroxide resin such as Amberlyst IRA-400 (OH). Whenever this methodology is performed with zwitterionic APIs [19,22] such as fluoroquinolones, it is required that the reaction proceeds in ammonia buffer solution so that complete ionisation can occur. The pure API–OSILs were isolated in very high yields (93–100%) after recrystallisation from chloroform/methanol mixtures. Scheme 1 shows the employed synthetic methodology.

### 2.2. Spectroscopic Characterisation

All products were characterised by ^1^H and ^13^C NMR and FTIR spectroscopic techniques, as well as elemental analysis.

In all cases, the ^1^H NMR spectra showed that the cation/anion proportion is strictly 1.0:1.0, in agreement with the intended stoichiometry (see Appendix A). In addition, only one set of signals was observed, meaning that the reactions were complete, and only one product was formed. No comparison with parent ciprofloxacin and norfloxacin is achievable due to a lack of solubility in the same solvent systems as the FQ–OSILs.

The FTIR spectra of the synthesised FQ–OSILs (Appendix A) show only discrete differences in characteristic signals in comparison with those from the parent fluoroquinolones. Typically, one would expect that the deprotonation by the cation hydroxides would occur at the carboxylic acid group forming a carboxylate group, which shows a very different wavenumber in comparison with the initial group. However, this observation was not possible because fluoroquinolones are zwitterions, and consequently, the carboxylic acid group is partially or totally in the form of carboxylate. In our sample of initial ciprofloxacin, the obtained FTIR spectrum showed complete ionisation of the carboxylic acid group, with only the signal from the stretching vibration of O=C-O^-^ appearing at 1589 cm^−1^ (see Figure 2A).

In this case, the FTIR spectra of the Cip–OSILs showed slight changes in the corresponding vibration band, more precisely between 1581 and 1575 cm^−1^ (e.g., 1575 cm^−1^ for [C_3_OMIM][Cip] in Figure 2B), thereby suggesting a change in the spatial vicinity of this group consistent with the formation of the salts. In the case of norfloxacin, the obtained FTIR spectrum (Figure 2C) showed incomplete ionisation of the carboxylic acid, and thus two bands were observable at 1727 and 1583 cm^−1^ from the stretching vibration of the carboxylic and carboxylate groups, respectively. The spectra of the Nor–OSILs showed complete disappearance of the signal from the former group, while the latter showed small changes to a range between 1583 and 1579 cm^−1^ (e.g., 1579 cm^−1^ for [C_3_OMIM][Nor] in Figure 2D). Furthermore, the 3600–3300 cm^−1^ zone always displays strong signals in the spectra of the final compounds, as opposed to the initial ones, which suggests the establishment of H-bonds between the cations and the anions.

### 2.3. Thermal Analysis

All prepared OSILs containing anionic ciprofloxacin and norfloxacin were studied by differential scanning calorimetry (DSC; see Appendix A). Table 1 contains the obtained data, namely, melting and glass transition temperatures, as well as the physical state at room temperature of the analysed compounds.

As expected, all compounds display lower melting temperatures than the parent fluoroquinolones. Globally, from the twelve FQ–OSILs, one half are organic salts ([Ch][Cip], [EMIM][Nor], [C_2_OHMIM][Cip], [C_2_OHDMIM][Cip], [C_2_OHDMIM][Nor] and [C_3_OMIM][Cip]), while the other half are considered ionic liquids ([Ch][Nor], [EMIM][Cip], [C_2_OHMIM][Nor], [C_3_OMIM][Nor], [C_16_Py][Cip] and [C_16_Py][Nor]) because their melting temperatures are, respectively, higher and lower than 100 °C. All solid compounds display only one melting temperature and are thus considered isomorphic salts. Moreover, the melting temperature range is comprehended between 92.9 and 197.5 °C, respectively, registered for [EMIM][Cip] and [C_2_OHDMIM][Cip], and was observed after one isotherm at 80 °C for 15 min, followed by a cooling cycle at 20 °C/min until −90 °C. In the cooling cycles following the melting phenomena, the OSILs do not display crystallization temperatures, meaning that they become supercooled salts, i.e., amorphous, after the first melt. This thermal behaviour is consistent with Type II, as described by Domínguez et al. [46].

From the referred set of ionic liquids, [C_2_OHMIM][Nor], [C_3_OMIM][Nor], [C_16_Py][Cip] and [C_16_Py][Nor] were obtained as room temperature ionic liquids (RTILs), consistent with the observed glass transition temperatures (T_g_). Even though several cooling cycles at 10 and 20 °C/min until −90 °C were performed, no endothermic events ascribable to melting phenomena were observed for the RTILs, in agreement with Domínguez’s Type I classification [46]. Note that from the four RTILs, three are norfloxacin-based, while only one contains ciprofloxacin. In fact, from the whole set of prepared FQ–OSILs, the melting temperatures of the Nor-based OSILs are invariably lower than the corresponding ones containing Cip. This observation is in line with the T_m_ of both free fluoroquinolones, with Cip and Nor recording 322 and 217 °C, respectively. Such considerable difference could not be expected by simply observing both chemical structures, as they only differ at the *N*-alkyl substituent (cyclopropyl versus ethyl) at the quinolone moiety. As OSILs, the three-dimensional packing of the fluoroquinolones and additional specific interactions of these drugs with the selected ammonium, methylimidazolium and pyridinium counterions [47] may account for the observed thermal behaviour of the compounds. These observations will be further explored in future works, e.g., by X-ray diffraction techniques, and published accordingly.

### 2.4. Water Solubility Studies

The solubility of the prepared organic salts based on ciprofloxacin and norfloxacin in water was determined by adding an excess amount of the compound to a fixed mass of water. Briefly, vials were kept under vigorous stirring and controlled temperature, from which samples were collected at different time periods to ensure the equilibrium was reached. After centrifugation, Cip and Nor were quantified through UV–Vis spectroscopy techniques.

All synthesised FQ–OSILs present higher solubility in water than the parent fluoroquinolones, either as free bases or as hydrochloride salts (only for ciprofloxacin). Figure 3 and Appendix A show the data obtained at 25 and 37 °C.

On a first observation, it is easily perceived that there is an enhanced increase in water solubility of the ciprofloxacin-based OSILs in comparison with the ones containing norfloxacin. Additionally, there is a direct correlation of the solubility with the temperature.

In both Nor- and Cip-based families of OSILs, the ones with the [EMIM] cation were found to be the least soluble ones, yet increases of 47- and 246-fold at 25 °C and of 72- and 343-fold at 37 °C were, respectively, observed. Conversely, the most soluble OSILs were obtained through the combination of the APIs with [C_2_OHMIM], with enhancements between 219- and 897-times at 25 °C, and 266- and 1416-times at 37 °C, respectively, for the Nor- and Cip-containing OSILs.

The observed general tendency is in agreement with the hydrophilic character of the studied cations: [EMIM] < [Ch] < [C_2_OHDMIM] ≈ [C_3_OMIM] < [C_2_OHMIM]. This relative behaviour is in agreement with the previously observed behaviour for ampicillin-based OSILs [20]. On the one hand, the [EMIM] cation does not possess any polar group in its structure, thereby inhibiting its ability to perform strong H-bonding interactions with the anions. On the other hand, the introduction of a polar hydroxyl group at the end of the ethyl group linked to the imidazolium ring in [C_2_OHMIM] increased the water solubility by 3.6- and 4.7-times for the corresponding Cip- and Nor-based OSILs. Changing the methylimidazolium moiety by an ammonium group, where the positive charge is more spatially hindered, had a deleterious effect in the solubility of the salts, which decreased by 1.8- and 2.8-times for [Ch][Cip] and [Ch][Nor], respectively, in comparison with the corresponding [C_2_OHMIM] ones. With the introduction of a methyl group between the methylimidazolium nitrogen atoms ([C_2_OHDMIM]) or at the end of the hydroxyethyl side chain ([C_3_OMIM]), intermediate hydrophilic properties are attained, which directly correlates with the solubility in water of the corresponding Cip– and Nor–OSILs.

In general, the anionic fluoroquinolone-based OSILs display a diminished enhancement of the solubility in water in comparison with the cationic fluoroquinolone-based OSILs previously reported [23]. In the latter group, with the exception of [Nor][Gly], the OSILS presented water solubility values between ca. 200 mg/mL up to totally soluble. However, this solubility profile may be attributed to the higher hydrophilic character of the chosen anions in comparison with the set of cations used in this work, and not actually to the anionic or cationic state of the fluoroquinolones.

### 2.5. Critical Micelle Concentration

The surfactant properties of the ionic liquids formed by the combination of the [C_16_Py] cation with both fluoroquinolones ([C_16_Py][Cip] and [C_16_Py][Nor]) were studied through the calculation of the corresponding critical micelle concentrations (CMCs) via ionic conductivity measurements. This data may be particularly relevant in order to develop novel drug delivery systems [48] for these FQ–OSILs. Table 2 and Appendix A show the obtained data for both OSILs, as well as for [C_16_Py]Cl.

The CMC value obtained for [C_16_Py]Cl is in good agreement with the value found in the open literature [49,50]. In comparison with [C_16_Py]Cl, the calculated CMC values for [C_16_Py][Cip] and [C_16_Py][Nor] are two orders of magnitude lower and correspond to a decrease of 92% and 95%, respectively. Despite the fact that smaller CMC values are expected for salts with organic anions in comparison with inorganic ones, such a drastic decrease comes as a surprise. In our previous work on ampicillin–OSILs [20], the combination of the antibiotic with the [C_16_Py] cation yielded a CMC value of 0.444 mM, which corresponds to a decrease of 58%. When comparing the three systems, we conclude that the level of hydration of Cip and Nor is much lower than that of ampicillin, leading to easier adsorption of these drugs on the surface of [C_16_Py] micellar structures, thus decreasing the repulsion between the polar groups at much lower concentrations in water.

### 2.6. Cytotoxicity Studies

The toxicity of the prepared FQ–OSILs and corresponding starting materials, namely, starting fluoroquinolones and halide salts, on 3T3 mouse fibroblasts was determined. Figure 4 plots the relative 3T3 cell viability in the presence of the starting materials and the FQ–OSILs at 10 µM.

From the whole set of compounds, only the ones that contained the [C_16_Py] cation were found to reduce cell viability by more than 50% at the tested concentration. Therefore, dose–response assays were carried out for these samples. The obtained results are plotted in Figure 5.

The determined IC_50_ values for [C_16_Py]Cl, [C_16_Py][Cip] and [C_16_Py][Nor] were, respectively, 5.46, 4.72 and 5.17 µM. These values were expected due to the surfactant properties of the cation and are in agreement with previous works [17,51].

### 2.7. Antimicrobial Activity Studies

The antimicrobial activity (IC_50_) of the prepared FQ–OSILs and the corresponding starting materials was determined against Gram-negative (*Klebsiella pneumoniae*) and Gram-positive (*Staphylococcus aureus, Bacillus subtilis*) bacteria.

Figure 6 plots the relative growth of (A) *Klebsiella pneumoniae*, (B) *Staphylococcus aureus* and (C) *Bacillus subtilis* in the presence of the starting materials and the FQ–OSILs at 10 µM for 6 h.

Both free fluoroquinolones and all prepared FQ–OSILs inhibited more than 50% growth of all bacteria strains at the tested concentration (10 µM). From the set of starting halide cations, [C_16_Py]Cl was the only one that exerted an effective antimicrobial effect.

Upon performing dose–response assays for the active compounds (see Appendix A), the corresponding IC_50_ values were determined. Table 3 and Table 4 present the gathered data for Cip- and Nor-based OSILs (and the corresponding free fluoroquinolones), respectively. The data for the starting cation halides are presented in Appendix A.

Ciprofloxacin was found to be particularly active against *B. subtilis* (3.84 nM), followed by *S. aureus* (29.16 nM) and *K. pneumoniae* (196.5 nM), as expected.

The combination of this antibiotic with the selected cations has rendered very interesting results against these bacteria strains, mostly *K. pneumoniae*, as similarly observed in previous works by Tajber and co-workers [42,43]. More specifically, all Cip–OSILs showed increased activity against this Gram-negative bacteria strain, with IC_50_ values comprehended between 9.88 ([C_16_Py][Cip]) and 64.8 nM ([EMIM][Cip]), which correspond to a remarkable relative decrease of inhibitory concentrations (RDIC_50_) of 19.9 and 3.0, respectively. While the antimicrobial activity of the former OSIL may come from an additive mechanism—because the [C_16_Py] cation itself shows an IC_50_ value of 2.55 mM (see Appendix A)—the potency of the remaining OSILs appears to have a synergistic behaviour since none of the cations display activity against this strain (or any of the others). In spite of [C_16_Py][Cip] showing toxicity against fibroblasts (see above), its IC_50_ value against *K. pneumoniae* is ca. 500-times lower (4.72 µM vs. 9.88 nM), making this Cip–OSIL particularly interesting against this strain. Moreover, [C_16_Py][Cip] showed almost no activity against *S. aureus* and also a 40% decrease against the commensal *B. subtilis* in comparison with ciprofloxacin. Hence, these results clearly suggest that [C_16_Py][Cip] has a very specific activity against *K. pneumoniae* over the remaining tested strains.

Of note is that *B. subtilis* is part of the human gut microbial ecosystem, whose perturbation may lead to the proliferation of resistant bacterial pathogens [52]. In fact, all Cip–OSILs had particularly reduced activities (20–30%) against *B. subtilis* in comparison with ciprofloxacin.

Regarding the pathogenic *S. aureus*, another Gram-positive bacteria strain, only [EMIM][Cip] and [C_2_OHDMIM][Cip] showed a slight increase in antimicrobial activity, with RDIC_50_ values between 1.2 and 1.4.

In its turn, free norfloxacin shows that, similarly to ciprofloxacin, it is also more active against Gram-positive bacteria (*S. aureus* and *B. subtilis*) than Gram-negative (*K. pneumoniae*) strains, but to a smaller extent than its fluoroquinolone sibling. Table 4 shows this data, as well as the one observed for the Nor-based OSILs.

The most active Nor-based OSIL is, once again, the one containing the [C_16_Py] cation, which showed a relative increase of 11.4-times (IC_50_ = 10.84 nM) against *S. aureus* in comparison with the free norfloxacin. This result is quite peculiar, as the analogous salt based on ciprofloxacin had similar activity against the Gram-negative *K. pneumoniae* microbial strain. Against the latter, [C_16_Py][Nor] has shown only a 30% increase in activity, with an IC_50_ value of 202.9 nM, which is much higher than the one recorded on *S. aureus*. However, it negatively affects commensal *B. subtilis* to a higher extent than norfloxacin.

An alternative promising broad-spectrum formulation may be [C_2_OHDMIM][Nor], which showed a 60% increased activity against *S. aureus* and 40% against *K. pneumoniae*. Desirably, it also lost 20% activity against commensal *B. subtilis*.

## 3. Conclusions

The ciprofloxacin- and norfloxacin-based OSILs presented in this paper clearly demonstrate that, by a simple and effective chemical transformation of standard fluoroquinolone drugs, it is possible to enhance their bioavailability and mitigate their polymorphic profiles. These advantages, allied to the possibility to modulate their antimicrobial spectrum while maintaining low toxicity towards healthy cells, make these FQ–OSILs a very promising tool to prepare novel effective formulations for these drugs.

In general, the prepared ciprofloxacin-based OSILs had much more encouraging antimicrobial data in comparison with the norfloxacin ones, despite both sets having positive results. According to the data regarding the three tested bacteria strains (*K. pneumoniae*, *S. aureus* and *B. subtilis*), we postulate that it is possible to develop highly bioavailable tailored formulations of both fluoroquinolones for applications against one or more pathogenic microbes with low toxicity against healthy cells and commensal bacteria, depending on the chosen combination with organic cations. On the one hand, the [C_16_Py]-based OSILs containing ciprofloxacin and norfloxacin were found to be particularly selective against different bacteria strains, respectively, *K. pneumoniae* and *S. aureus*, both at subtoxic concentrations. Such distinct behaviour must surge from distinct interactions between the cation, the drugs and the cell external membrane/wall of the bacteria, and will be studied in the future. In addition, all formulations based on the [C_16_Py] cation assembled into micelles, with possible applications in novel drug delivery systems of the antibiotics.

Alternatively, highly water-soluble formulations with a selective spectrum of activity could consist of [Ch][Cip], which was shown to be ca. 3.5-times more active to this strain than to *S. aureus.* However, its IC_50_ values against *K. pneumoniae* are some of the lowest recorded (51.12 nM). Hence, promising alternative formulations can consist of [C_2_OHMIM]- and [C_3_OMIM][Cip], which demonstrated selectivity of ca. 1.7-times towards the same strain, with IC_50_ values of 36.43 and 47.61 nM, respectively. Moreover, both latter compounds are more water-soluble than [Ch][Cip] and probably more bioavailable.

On the other hand, if a broad-spectrum antibiotic ionic formulation is sought, its combination with [EMIM] or [C_2_OHDMIM] is promising as they are highly active against both Gram-negative and Gram-positive bacteria strains, in particular *K. pneumoniae* and *S. aureus*, respectively.

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
