# Peer review of "Antimicrobial Activities of Highly Bioavailable Organic Salts and Ionic Liquids from Fluoroquinolones"

_pharmaceutics, 2020, doi:10.3390/pharmaceutics12080694_

Round 1
Reviewer 1 Report
The article entitled “Highly bioavailable Organic Salts and Ionic Liquids from fluoroquinolones presenting tailorable antimicrobial activities against Gram-positive and Gram-negative bacteria” presents the synthesis and the study of some physico-chemical properties, and toxicological studies of a series of organic salts and ionic liquids (OSILs) based on the active pharmaceutical ingredients (API) ciprofloxacin and norfloxacin acting as anions. The cations included various structures commonly used in ionic liquids compositions. Regarding the scientific merit this work represents an advance in the API-OSILs field. I recommend this article to be accepted to publication with some major changes. These changes aim to add results and discussion that would definitely improve the article. The changes are described below:
- Figure 1
- Where it is written [C3OHMIM]Cl should be [C3OMIM]Cl.
- Table 1
- Specify the temperature of the “physical state” description.
- Thermal properties:
- The DSC curves (in the SI) should be clarified, one can tell what is going on chronologically in the analysis only by looking at the curves. For example: Which cycle is the first and what was done next?
- For several of the compounds there is some transition above 150 °C that could be related to thermal decomposition. Thermogravimetric (TGA) analysis must be carried out to investigate these transitions. Some of the authors have a published work (International Journal of Pharmaceutics 469 (2014) 179–189) where the TGA measurements show that some organic salts with ciprofloxacin and norfloxacin as cations undergo thermal decomposition at relatively low temperatures.
- Considering the work mentioned above it is important to make a comparison of the thermal properties of the organic salts with ciprofloxacin and norfloxacin acting as cations (in the cited article) or anions (present work).
- CMC determination:
- The authors should tell, in the discussion section, the experimental technique used to determine the cmc. Currently this information in available only in the SI.
- In the SI should be show the conductivity curves and the methodology used to exactly determine the point of cmc.
- Antimicrobial activity studies
- Although the authors say (in the Conclusion) that future studies on the interactions between cations, drugs and cell membranes will be carried out, some hypothesis can be suggested in the present work. This would enrich the discussion of the results from a more chemical point of view.
Author Response
Thank you for your revision.
The article entitled “Highly bioavailable Organic Salts and Ionic Liquids from fluoroquinolones presenting tailorable antimicrobial activities against Gram-positive and Gram-negative bacteria” presents the synthesis and the study of some physicochemical properties, and toxicological studies of a series of organic salts and ionic liquids (OSILs) based on the active pharmaceutical ingredients (API) ciprofloxacin and norfloxacin acting as anions. The cations included various structures commonly used in ionic liquids compositions. Regarding the scientific merit this work represents an advance in the API-OSILs field. I recommend this article to be accepted to publication with some major changes. These changes aim to add results and discussion that would definitely improve the article. The main changes are described below:
- Figure 1. Where it is written [C3OHMIM]Cl should be [C3OMIM]Cl.
Authors: Thank you for noticing this typo. It is now corrected.
- Table 1. Specify the temperature of the “physical state” description.
Authors: Table’s caption now describes “Physical state at room temperature”
- Thermal properties: the DSC curves (in the SI) should be clarified, one can tell what is going on chronologically in the analysis only by looking at the curves. For example: Which cycle is the first and what was done next?
Authors: A color-coded caption for each DSC has been added to the thermograms.
- For several of the compounds there is some transition above 150 °C that could be related to thermal decomposition. Thermogravimetric (TGA) analysis must be carried out to investigate these transitions. Some of the authors have a published work (International Journal of Pharmaceutics 469 (2014) 179–189) where the TGA measurements show that some organic salts with ciprofloxacin and norfloxacin as cations undergo thermal decomposition at relatively low temperatures. Considering the work mentioned above it is important to make a comparison of the thermal properties of the organic salts with ciprofloxacin and norfloxacin acting as cations (in the cited article) or anions (present work).
Authors: All prepared API-ILs were characterized by DSC in order to illustrate their thermal properties. In the case of API-ILs is important to identify the presence of glass transition temperatures (Tg), melting points as well as polymorphs. It seems relevant to check the crystalline or amorphous behaviour and not the TGA measurements. In our opinion, the detailed thermal study by DSC is more important than TGA according to further pharmaceutical applications.
- CMC determination: The authors should tell, in the discussion section, the experimental technique used to determine the CMC. Currently this information in available only in the SI.
Authors: This information is now provided in the discussion section (line 241).
- In the SI should be show the conductivity curves and the methodology used to exactly determine the point of CMC.
Authors: This information is now provided in the SI.
- Antimicrobial activity studies: Although the authors say (in the Conclusion) that future studies on the interactions between cations, drugs and cell membranes will be carried out, some hypothesis can be suggested in the present work. This would enrich the discussion of the results from a more chemical point of view.
Authors: Different studies can be considered in order to elucidate the cation-anion interactions including the use of liposomes or computational studies. In our opinion the brief indication about future studies is appropriate in the present work.
Reviewer 2 Report
The manuscript provides a rather comprehensive characterization of twelve novel API-ILs with antibacterial activity. The authors have an impressive experience in such studies, and I suggest publishing the work after resolving several minor issues.
- I advise the authors to provide more discussion on possible advantages of the newly synthesized API-ILs in comparison with cetylpyridinium chloride.
- What are error bars in Figure 3?
- Figure 5 should be supplied with the analysis of statistical differences between the original drugs and the API-ILs.
- The IC50 value for [C16Py][Cip] in S. aureus given in Table 3 seems to contradict the activity of this substance shown in Figure 5.
Author Response
Thank you for your revision.
The manuscript provides a rather comprehensive characterization of twelve novel API-ILs with antibacterial activity. The authors have an impressive experience in such studies, and I suggest publishing the work after resolving several minor issues.
- I advise the authors to provide more discussion on possible advantages of the newly synthesized API-ILs in comparison with cetylpyridinium chloride.
Authors: The new API-ILs can improve different properties such as solubility, permeability, avoid polymorphism as well as more effective anti-bacterial activities. It is clear that Cetylpyridinium as cation and API as anion is different than cetylpyridinium chloride combined with API.
- What are error bars in Figure 3?
Authors: The sentence “Values represent mean ± standard error of the mean (SEM) of at least three independent experiments” was added to the Figure’s caption.
- Figure 5 should be supplied with the analysis of statistical differences between the original drugs and the API-ILs.
Authors: This analysis has been added to Figure 6 (previously Figure 5).
- The IC50 value for [C16Py][Cip] in aureus given in Table 3 seems to contradict the activity of this substance shown in Figure 5.
Authors: Respectfully, it appears that the reviewer may have misread the data on the graphics. At maximum concentration (10 micromolar – shown in Figure 6) there is 84% decrease in S. aureus growth, so an IC50 value of 1.43 micromolar (1430 nanomolar as shown in Table 3) is plausible, and does not contradict any determined value. The following graphs show the S. aureus growth at the different tested [C16Py][Cip] concentrations:
see figure in the attach word document

Reviewer 3 Report
The work entitled “Antimicrobial Activities of Highly Bioavailable Organic Salts and Ionic Liquids from Fluoroquinolones” describes the preparation and characterization of twelve Organic Salts and Ionic Liquids (OSILs) with ciprofloxacin and norfloxacin as anions obtained by means of a buffer-assisted neutralization methodology. The solubility, toxicity towards healthy cells, and antimicrobial activity against Klebsiella pneumoniae were evaluated. The approach is similar to previous works reported by the same authors, but the obtained ILs are well characterized and the results are of high interest. For this reason, I recommend the manuscript for publication after the following minor revisions are made and relevant literature is added.
Numbering of tables and figures is confusing. For example, Figure 1 does not present any captions, there are two figures “3” and I would guess that Table 5 is actually Table 4.
In line 53 and 55 the temperatures were reported in Kelvin but in other parts of the text temperature are in Celsius. Please correct these inconsistencies.
In the DSC part, the authors describe the thermal behavior of the OSILs. These behaviors are quite common for ionic liquids and ionic salts. The thermal behavior of ionic liquids has been classified into three different classes (Gómez E, Calvarn N, Domínguez A. Thermal behavior of pure ionic liquids Scott handy. Intechopen. 2015, 8, 199–228, Guazzelli et al. Journal of Thermal Analysis and Calorimetry (2019) 138, 3335–3345). I suggest to evaluating the thermal behavior of OSILs and classify them in accordance to the above mentioned classification.
In the same DSC part, the authors compared the melting temperatures of OSILs with two different anions (Cip and Nor). The analysis is well structured and the trend of melting points of OSILs follows that of their parent compounds. On the other hand, the trend as a function of cations has not been commented in depth. The effect of cation is widely studied in literature (J. Phys. Chem. B 2010, 114, 3601–3607, Frontiers in Chemistry 2019, 7, 450, Rooney D., Jacquemin J., Gardas R. (2009) Thermophysical Properties of Ionic Liquids. In: Kirchner B. (eds) Ionic Liquids. Topics in Current Chemistry, vol 290. Springer, Berlin, Heidelberg). A comparison of the melting temperatures of EMIM, C2OHMIM, and C3OHMIM based ionic liquids was reported by Mezzetta et al. (Journal of Molecular Liquids 289 (2019) 111155).
The melting temperature of [C2OHDMIM][Cip] is incorrect. In Figure S42 the DSC thermogram of [C2OHDMIM][Cip] is not complete. I suggest to repeat the analysis up to 220-230 °C.
In the water solubility section, the influence of the structure on solubility is well described. However, the increase in OSILs solubility with respectto their parent compounds must be evaluated following the guide line of the United States Pharmacopeia, British Pharmacopoeia (International Scholarly Research Network ISRN Pharmaceutics Volume 2012, Article ID 195727, 10 pages) or European Pharmacopoeia (Ph. Eur.).
In the critical micelle concentration (CMC) section, the CMC obtained for [C16Py][Cl] should be compared also with more recent works (i.e. J. Phys. Chem. B 2001, 105, 51, 12823–12831 and J. Phys. Chem. B 2017, 121, 8742−8755).
Author Response
Thank your for your revision.
The work entitled “Antimicrobial Activities of Highly Bioavailable Organic Salts and Ionic Liquids from Fluoroquinolones” describes the preparation and characterization of twelve Organic Salts and Ionic Liquids (OSILs) with ciprofloxacin and norfloxacin as anions obtained by means of a buffer-assisted neutralization methodology. The solubility, toxicity towards healthy cells, and antimicrobial activity against Klebsiella pneumoniae were evaluated. The approach is similar to previous works reported by the same authors, but the obtained ILs are well characterized and the results are of high interest. For this reason, I recommend the manuscript for publication after the following minor revisions are made and relevant literature is added.
- Numbering of tables and figures is confusing. For example, Figure 1 does not present any captions, there are two figures “3” and I would guess that Table 5 is actually Table 4.
Authors: We appreciate the remarks, and the numbering is now corrected.
- In lines 53 and 55 the temperatures were reported in Kelvin but in other parts of the text temperature are in Celsius. Please correct these inconsistencies.
Authors: These inconsistencies are now corrected.
- In the DSC part, the authors describe the thermal behavior of the OSILs. These behaviors are quite common for ionic liquids and ionic salts. The thermal behavior of ionic liquids has been classified into three different classes (Gómez E, Calvarn N, Domínguez A. Thermal behavior of pure ionic liquids Scott handy. Intechopen. 2015, 8, 199–228, Guazzelli et al. Journal of Thermal Analysis and Calorimetry (2019) 138, 3335–3345). I suggest to evaluating the thermal behavior of OSILs and classify them in accordance to the above mentioned classification.
Authors: appreciate the suggestion. We added this information in lines 176 and 182 (page 6) of the manuscript and added reference 46.
- In the same DSC part, the authors compared the melting temperatures of OSILs with two different anions (Cip and Nor). The analysis is well structured and the trend of melting points of OSILs follows that of their parent compounds. On the other hand, the trend as a function of cations has not been commented in depth. The effect of cation is widely studied in literature (J. Phys. Chem. B 2010, 114, 3601–3607, Frontiers in Chemistry 2019, 7, 450, Rooney D., Jacquemin J., Gardas R. (2009) Thermophysical Properties of Ionic Liquids. In: Kirchner B. (eds) Ionic Liquids. Topics in Current Chemistry, vol 290. Springer, Berlin, Heidelberg). A comparison of the melting temperatures of EMIM, C2OHMIM, and C3OMIM based ionic liquids was reported by Mezzetta et al. (Journal of Molecular Liquids 289 (2019) 111155).
Authors: In this context, it is important to check the anion effect because the main interest in the thermal analysis is related to polymorphic behaviour. In our opinion, the comparison between original APIs (Cip or Nor) and the new API-ILs is the relevant point for discussion.
- The melting temperature of [C2OHDMIM][Cip] is incorrect. In Figure S42 the DSC thermogram of [C2OHDMIM][Cip] is not complete. I suggest to repeat the analysis up to 220-230 °C.
Authors: The DSC thermogram of [C2OHDMIM][Cip] has been repeated and is provided as Figure S42 in SI. The observed melting temperature has been updated in Table 1.
- In the water solubility section, the influence of the structure on solubility is well described. However, the increase in OSILs solubility with respect to their parent compounds must be evaluated following the guide line of the United States Pharmacopeia, British Pharmacopoeia (International Scholarly Research Network ISRN Pharmaceutics Volume 2012, Article ID 195727, 10 pages) or European Pharmacopoeia (Ph. Eur.).
Authors: As requested, a couple of lines were introduced in the Introduction (Lines 56-59) mentioning that the formation of salts using active pharmaceutical ingredients is a well-known chemical modification technique.
- In the critical micelle concentration (CMC) section, the CMC obtained for [C16Py][Cl] should be compared also with more recent works (i.e. J. Phys. Chem. B 2001, 105, 51, 12823–12831 and J. Phys. Chem. B 2017, 121, 8742−8755).
Authors: The value of the CMC obtained for [C16Py][Cl] was compared with that of J. Phys. Chem. B 2001, 105, 51, 12823–12831 (now reference 51) in Table 2 and lines 246-247. The article J. Phys. Chem. B 2017, 121, 8742−8755 does not measure the CMC of [C16Py][Cl] but also uses the 2001 previously referred article.